Methods

# Generation of marmoset primordial germ cell–like cells under chemically defined conditions

Julia Kurlovich[1,*], Ignacio Rodriguez Polo[1,2,3,*] , Oleksandr Dovgusha[1], Yuliia Tereshchenko[2,4], Carmela Rieline V Cruz[1], Rüdiger Behr[2,4] , Ufuk Günesdogan[1,5] 

**Primordial germ cells (PGCs) are the embryonic precursors of sperm and oocytes, which transmit genetic/epigenetic information across generations. Mouse PGC and subsequent gamete development can be fully reconstituted in vitro, opening up new avenues for germ cell studies in biomedical research. However, PGCs show molecular differences between rodents and humans. Therefore, to establish an in vitro system that is closely related to humans, we studied PGC development in vivo and in vitro in the common marmoset monkey *Callithrix jacchus* (*cj*). Gonadal cjPGCs at embryonic day 74 express SOX17, AP2ɣ, BLIMP1, NANOG, and OCT4A, which is reminiscent of human PGCs. We established transgene-free induced pluripotent stem cell (cjiPSC) lines from foetal and postnatal fibroblasts. These cjiPSCs, cultured in defined and feeder-free conditions, can be differentiated into precursors of mesendoderm and subsequently into cjPGC-like cells (cjPGCLCs) with a transcriptome similar to human PGCs/PGCLCs. Our results not only pave the way for studying PGC development in a non-human primate in vitro under experimentally controlled conditions, but also provide the opportunity to derive functional marmoset gametes in future studies.**

## Introduction

Mammalian primordial germ cells (PGCs) are specified during or shortly after implantation of the embryo in the uterus and give rise to gametes after birth. During specification, PGCs up-regulate a set of key transcription factors, whose activity leads to suppression of somatic differentiation, reacquisition of transient pluripotency, and epigenetic programming, including global DNA demethylation (Kobayashi & Surani, 2018). PGCs then migrate to the prospective gonads, where they eventually undergo sex determination and gametogenesis.

The development of post-implantation embryos and PGCs in rodents shows significant differences compared with other mammals (Kobayashi & Surani, 2018). PGC specification within the cup-shaped epiblast of mice occurs after implantation through secreted BMP4 signalling from the adjacent extraembryonic ectoderm (ExE) and WNT3A activity within the epiblast and surrounding visceral endoderm (Lawson et al, 1999; Ohinata et al, 2009; Aramaki et al, 2013). In particular, the downstream WNT effector BRACHYURY (also known as T or TBX-T) induces the expression of the key PGC transcription factors BLIMP1 (encoded by *Prdm1*) and PRDM14, which results in the expression of AP2ɣ (encoded by *Tfap2c*) (Aramaki et al, 2013). These factors are required for PGC development, as they control the transcriptional programme of PGCs, which involves the maintenance or up-regulation of pluripotency factors, including OCT4A, SOX2, and NANOG (Ohinata et al, 2005; Yamaji et al, 2008; Weber et al, 2010). Accordingly, the induced expression of BLIMP1, PRDM14, and AP2ɣ is sufficient to drive the PGC fate in vitro (Magnúsdóttir et al, 2013; Nakaki et al, 2013).

In contrast to rodents, non-human primate (NHP) and human post-implantation embryos consist of a bilaminar disc with the epiblast overlying the hypoblast (primitive endoderm), and a structure reminiscent of the mouse ExE has not been identified to date. Primate PGC specification relies on BMP and WNT activity acting through GATA2/3 and EOMES, respectively (Kojima et al, 2017, 2021). This results in the induction of a core transcriptional network including SOX17, BLIMP1, and AP2ɣ (Irie et al, 2015; Sasaki et al, 2015; Sugawa et al, 2015). Before gastrulation, PGCs in cynomolgus monkeys and marmosets are located within the extraembryonic amnion, indicating an extraembryonic origin of NHP PGCs (Sasaki et al, 2016; Bergmann et al, 2022). However, it has been suggested that both amnion and PGCs might arise from common epiblast progenitor cells characterised by the expression of AP2α (encoded by *TFAP2A*) (Chen et al, 2019; Castillo-Venzor et al, 2023). After PGC specification, human and macaque PGCs up-regulate pluripotency-associated genes such as NANOG and OCT4A, but in contrast to rodents, SOX2 remains repressed (Irie et al, 2015; Sasaki et al, 2015; Sugawa et al, 2015).

[1]Göttingen Center for Molecular Biosciences, Department of Developmental Biology, University of Göttingen, Göttingen, Germany  [2]German Primate Center—Leibniz Institute for Primate Research, Research Platform Degenerative Diseases, Göttingen, Germany  [3]Stem Cell and Human Development Laboratory, The Francis Crick Institute, London, UK  [4]German Center for Cardiovascular Research (DZHK), Partner Site Göttingen, Göttingen, Germany  [5]Department for Molecular Developmental Biology, Max Planck Institute for Multidisciplinary Sciences, Göttingen, Germany

Correspondence: rbehr@dpz.eu; ufuk.guenesdogan@biologie.uni-goettingen.de
*Julia Kurlovich and Ignacio Rodriguez Polo contributed equally to this work

The developmental pathway of mammalian germ cells can be reconstituted in vitro. In particular, pluripotent stem cells (PSCs) such as embryonic stem cells (ESCs) or induced pluripotent stem cells (iPSCs) of rodents, primates, and other mammals can be differentiated into PGC-like cells (PGCLCs), which represent early PGCs in vivo (Hayashi et al, 2011; Irie et al, 2015; Sasaki et al, 2015; Sugawa et al, 2015; Sosa et al, 2018; Overeem et al, 2023; Seita et al, 2023; Shono et al, 2023). Mouse PGCLCs (mPGCLCs) can even give rise to functional mature oocytes or spermatocytes in vitro when aggregated with gonadal somatic cells to generate reconstituted ovaries or testes (Hayashi et al, 2012; Hikabe et al, 2016; Ishikura et al, 2016, 2021; Yoshino et al, 2021). In addition, human or NHP PGCLCs aggregated with mouse gonadal cells or transplanted into the gonadal niche can further differentiate and give rise to late PGCLCs, pre-meiotic spermatogonia-/oogonia-like cells, or meiotic oocytes (Yamashiro et al, 2018; Hwang et al, 2020; Gyobu-Motani et al, 2023; Seita et al, 2023; Shono et al, 2023).

The differentiation of PGCLCs depends on the pluripotent state of iPSCs or ESCs. For example, human ESCs (hESCs) cultured with four inhibitors and LIF, bFGF, and TGFβ harbour some features of naïve pluripotency, including developmental competence to directly acquire the PGC fate upon the addition of BMP2/4 and other cytokines (Irie et al, 2015). In contrast, human or macaque ESCs cultured in other commercially available media, such as Essential 8 (E8), exhibit primed pluripotency and give rise to PGCs in response to the addition of BMP2/4 only after differentiation into precursors of mesendoderm-like cells (pre-ME) (Kobayashi et al, 2017).

These studies open up the prospect of complete in vitro gametogenesis using human pluripotent stem cells (hPSCs). However, monitoring the faithful in vitro differentiation of human gametes requires functional experiments and a reference to embryonic germ cells, which is not possible for ethical reasons. Therefore, it is necessary to establish alternative model systems for germ cell research, which are closely related to humans. We focussed on the marmoset *Callithrix jacchus* (cj), a small New World monkey, which has a short generation time compared with other NHPs, a larger litter size (two to three pups instead of singletons in macaques), and a higher gestation frequency (2 litters per year instead of one every 2–3 yr in macaques) (Mansfield, 2003; Pittet et al, 2017). In addition, in vitro fertilisation and transgenesis technologies are well established for this species (Sasaki et al, 2009; Okano et al, 2012; Takahashi et al, 2014; Drummer et al, 2021). We show that post-migratory PGCs in marmoset gonads express PGC genes known to be expressed also in human PGCs (hPGCs). We established and characterised marmoset iPSC (cjiPSC) lines under transgene- and feeder-free conditions. Notably, these cjiPSC lines were generated and maintained following previously established conditions for human, macaque, and baboon cells (Stauske et al, 2020; Yoshimatsu et al, 2021; Rodríguez-Polo et al, 2022). Importantly, under specific chemically defined culture conditions, these cells can be induced into marmoset PGCLCs (cjPGCLCs), which exhibit a similar expression profile to embryonic cjPGCs and hPGCs/human PGCLCs (hPGCLCs). Thus, our approach establishes the first step of marmoset in vitro gametogenesis under defined conditions using PSCs.

# Results

## Characterisation of gonadal marmoset PGCs

We asked whether post-migratory cjPGCs within the gonads have an expression profile similar to that of other NHPs and humans. We isolated genital ridges from two male marmoset embryos at embryonic day (E) 74, which corresponds to Carnegie stage (CS) 18 (Butler, 2017). CjPGCs were readily identified in the developing gonads by the co-expression of the characteristic PGC markers AP2ɣ and BLIMP1, as well as pluripotency factors OCT4A and NANOG (Fig 1). Importantly, cjPGCs were positive for SOX17 and negative for SOX2. These results suggest that cjPGCs have an expression profile of key PGC markers comparable to other primate species, including humans and macaques (Irie et al, 2015; Sasaki et al, 2015, 2016; Sugawa et al, 2015; Kobayashi et al, 2017).

## Marmoset fibroblast reprogramming into cjiPSCs

To further establish the marmoset monkey as a model for studying germ cell development in vitro, we first generated a panel of cjiPSC lines. We followed our previously described approach to derive iPSCs from human, macaque, and baboon fibroblasts under feeder- and transgene-free conditions (Stauske et al, 2020; Rodríguez-Polo et al, 2022) (Fig 2A). To this end, primary fibroblasts isolated from one foetal and two postnatal marmosets were transfected with episomal plasmids for the transient expression of human OCT3/4, SOX2, KLF4, L-MYC, LIN28, and a small hairpin RNA against p53 (Okita et al, 2011). After transfection with the episomal plasmids, primary cells were maintained for ~40 d in E8 medium or Universal Primate Pluripotent Stem Cell (UPPS) medium (Table S1) (Rodríguez-Polo et al, 2022), resulting in putative cjiPSC colonies with characteristic morphology (Fig 2A and B).

Between d30 and d40, cjiPSC colonies were manually picked and expanded in UPPS medium. After ~5 passages, the cjiPSC lines gave rise to flat and compact colonies with sharp borders and cells with an apparently high nucleus-to-cytoplasm ratio, which is reminiscent of primed hiPSCs (Stauske et al, 2020). Between passages 5 and 10, we selected six cjiPSC lines, two derived from neonatal fibroblasts (DPZ_cjiPSC#1 and DPZ_cjiPSC#6) and four from foetal fibroblasts (DPZ_cjiPSC#2-5) (Fig 2C). However, in contrast to human or other NHP iPSCs under identical conditions (Stauske et al, 2020), the putative cjiPSC lines were less stable at early passages resulting in a high percentage of differentiated cells. To address this, we empirically tested the addition of different combinations of small molecules and cytokines to the UPPS medium. We were able to stabilise the cjiPSC lines by alternating the cell culture conditions every two to three passages between UPPS medium alone and UPPS medium together with Activin A (ActA) and LIF (Fig 2A). Around passages 10–15, six selected cjiPSC lines (DPZ_cjiPSC#1-6) could be stably maintained in UPPS medium alone (Fig 2C). After expansion, we confirmed pluripotency-associated alkaline phosphatase activity in the six cjiPSC lines (Fig 2D).

PCR analysis showed that five of six lines (DPZ_cjiPSC#2-6) lost the episomes by passage 15 (Fig 3A). In addition, semi-quantitative transcript abundance analysis by RT–PCR confirmed that the cell

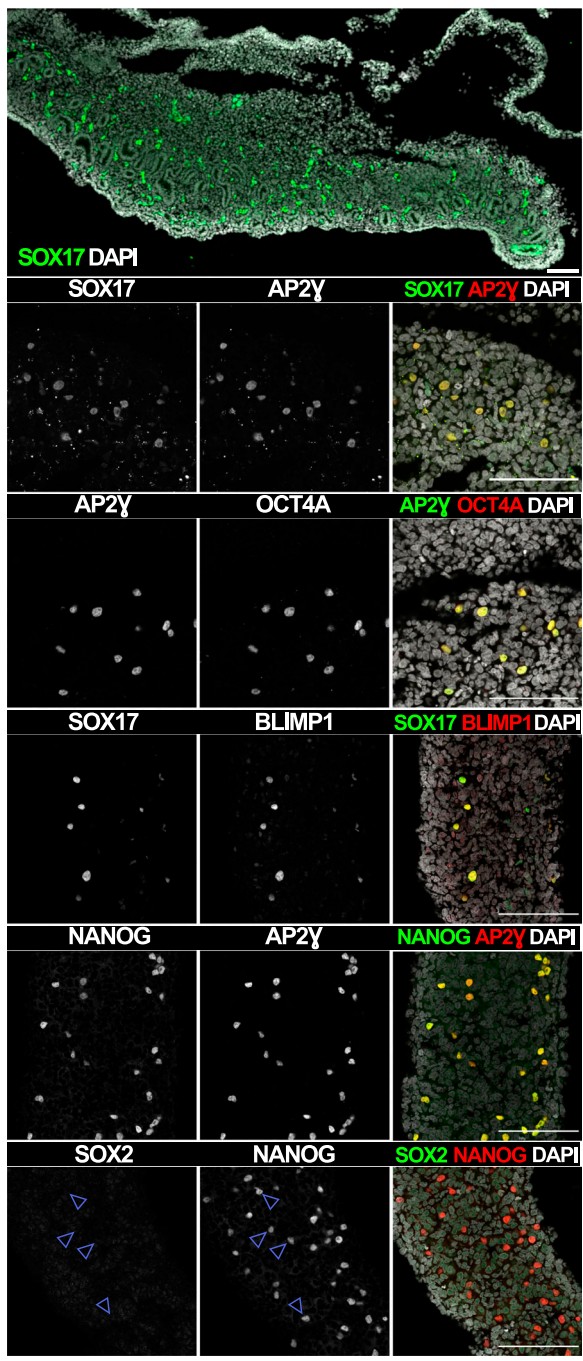

**Figure 1. cjPGCs at E74 show an expression profile similar to hPGCs.**
Immunofluorescence images of marmoset E74 genital ridge sections, which were stained for SOX17, AP2γ, OCT4A, NANOG, BLIMP, and SOX2. Top three rows: genital ridges isolated from embryo 1; bottom three rows: genital ridges isolated from embryo 2. Top row: scale bar, 50 μm. Other rows: scale bars, 100 μm.

lines expressed the endogenous pluripotency factors *OCT4*, *KLF4*, and *c-MYC* at high levels (Fig 3B). In addition, four further characterised cjiPSC lines (DPZ_cjiPSC#1-3, 5) showed the protein expression of pluripotency factors OCT4A, LIN28, NANOG, and SOX2, as well as the pluripotency-associated glycans TRA-1-81 and TRA-1-60 (Fig 3C). Finally, we tested the differentiation potential of these

cjiPSC lines using an embryoid body (EB) formation assay. Cell aggregates were generated and exposed to differentiation medium (see the Materials and Methods section) for 8d in suspension followed by 17d culture after EB attachment to gelatine-coated coverslips. Immunofluorescence stainings confirmed that the outgrowths of cell aggregates developed into representative cell types of the three embryonic germ layers, as judged by the expression of smooth muscle actin (mesoderm), α-fetoprotein (endoderm), and β-III-tubulin (ectoderm) (Fig 3D). Taken together, we have established three stable, transgene- and feeder-free pluripotent cjiPSC lines (DPZ_cjiPSC#2, DPZ_cjiPSC#3, and DPZ_cjiPSC#5).

**Differentiation of cjiPSCs towards cjPGCLCs**

To generate cjPGCLCs from cjiPSCs, we tested whether cjiPSCs (DPZ_cjiPSC#2, DPZ_cjiPSC#3, and DPZ_cjiPSC#5) are competent to be directly induced into cjPGCLCs. CjiPSCs were adapted to a modified UPPS medium containing forskolin, LIF, and a low concentration of ActA, which improves the stability of cjiPSCs (Petkov et al, 2020). Dissociated cjiPSCs were reaggregated in ultra-low attachment wells to allow the formation of EBs in the presence of BMP4, EGF, SCF, and LIF. At d2 of differentiation, small clusters of AP2γ-SOX17 and/or AP2γ-BLIMP1 double-positive cells were detected (Fig S1A). However, by d6 of differentiation AP2γ-SOX17 and/or AP2γ-BLIMP1 expression appeared to be mutually exclusive, suggesting that the cells adopted different cell fates during the time course of differentiation (Fig S1B).

Next, we tested a two-step protocol that has been successfully applied to differentiate macaque and human PGCLCs from PSCs (Kobayashi et al, 2017). In this approach, PSCs are differentiated into pre-ME using ActA and CHIR, a small molecule inhibitor of GSK3, before being induced into PGCLCs by the addition of BMP4, EGF, SCF, and LIF. To this end, cjiPSCs (DPZ_cjiPSC#2 unless otherwise stated) or cynomolgus monkey ESCs (cyESCs), used as a positive control, were adapted to E8 medium and cultured on mouse embryonic fibroblasts. At d4 of differentiation, cynomolgus monkey EBs gave rise to large clusters of SOX17-AP2γ– and NANOG-BLIMP1–positive cells (Fig S1C), consistent with a previous study (Kobayashi et al, 2017). In contrast, although a considerable number of cells within marmoset EBs expressed SOX17, only very few cells co-expressed AP2γ (Fig S1D). Furthermore, NANOG and BLIMP1 expression was barely detectable in marmoset EBs, indicating that SOX17-positive cells may have acquired an endodermal rather than the PGC fate. These results suggest that cjiPSCs require different maintenance or differentiation conditions for the PGCLC fate than other primate pluripotent cells.

Next, we adapted cjiPSCs (DPZ_cjiPSC#2, DPZ_cjiPSC#3, and DPZ_cjiPSC#5) to TeSR-E8 (TESR) medium without feeders, which is regularly used to maintain human iPSCs (hiPSCs). cjiPSCs cultured in TESR medium retained undifferentiated morphology and expressed the pluripotency factor OCT4A (Fig S2A). However, only the differentiation of hiPSCs but not cjiPSCs into pre-ME, followed by induction into PGCLCs, resulted in large clusters of SOX17-AP2γ– and NANOG-BLIMP1–positive cells (Fig S2B and C).

The culture of cjiPSCs in TESR medium did not result in successful cjPGCLC differentiation. Thus, we considered the origin of

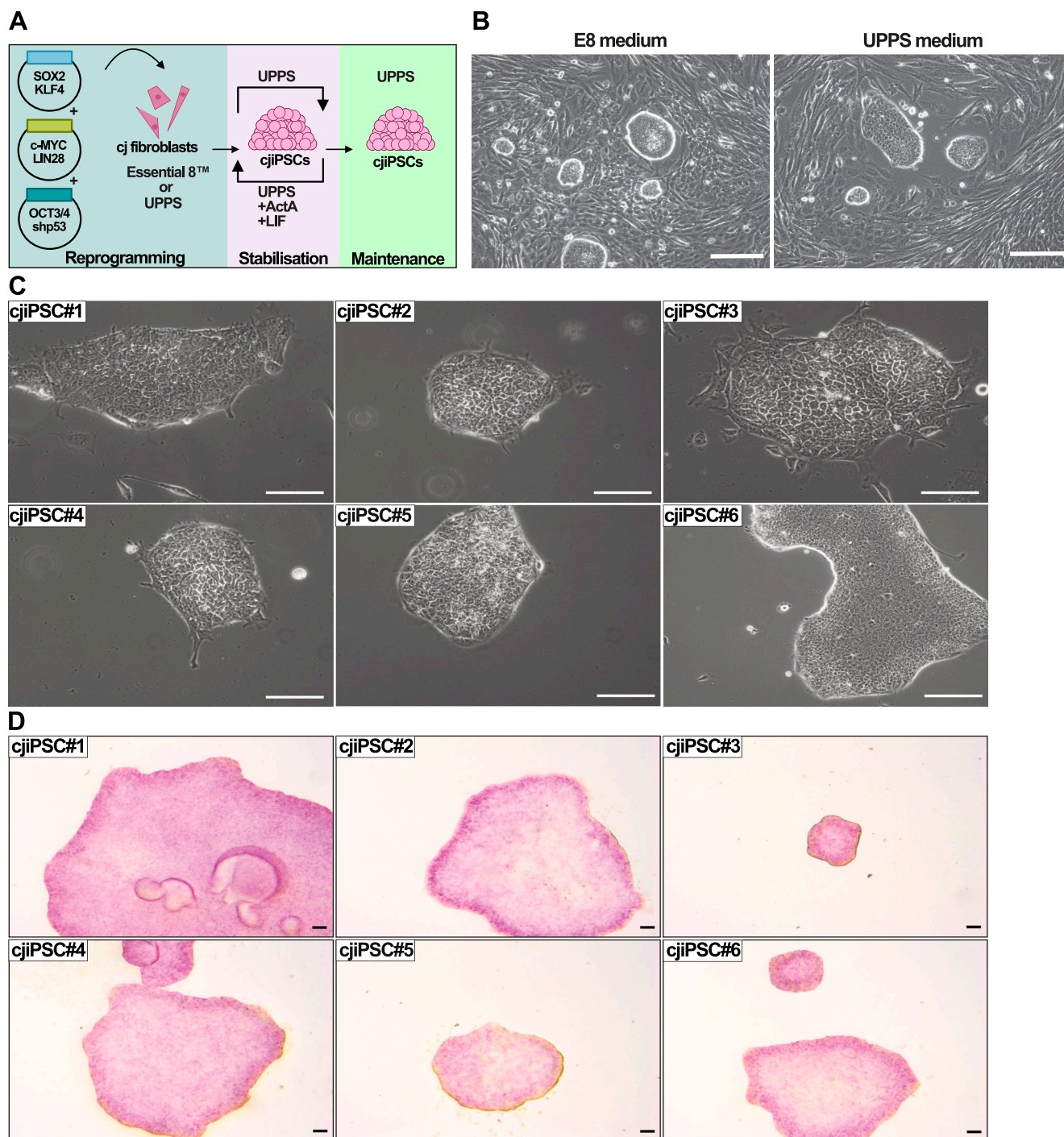

**Figure 2. Derivation and characterisation of cjiPSCs.**
**(A)** Overview of cjiPSC derivation by reprogramming of fibroblasts. Culture conditions for cjiPSC stabilisation and maintenance conditions are depicted. **(B)** Brightfield images of foetal fibroblast primary culture 30d after transfection with reprogramming plasmids maintained in E8 medium (left) or Universal Primate Pluripotent Stem Cell medium (right). **(C)** Brightfield images of cjiPSC colonies (DPZ_cjiPSC#1-6) maintained in Universal Primate Pluripotent Stem Cell medium. **(D)** Alkaline phosphatase staining of cjiPSC colonies (DPZ_cjiPSC#1-6). Scale bars, 100 μm.

primate PGCs, which may be derived from or have the same precursors as extraembryonic amniotic cells (Sasaki et al, 2016; Chen et al, 2019; Castillo-Venzor et al, 2023). Because hESCs or hiPSCs in human expanded potential stem cell medium (hEPSCM, Table S1)

have the developmental potential to develop into embryonic germ layers, hPGCLCs, and extraembryonic lineages (Gao et al, 2019), we sought to adapt cjiPSCs to hEPSCM. Although the transfer of cjiPSCs from TESR to hEPSCM resulted in cell death, cjiPSCs could be

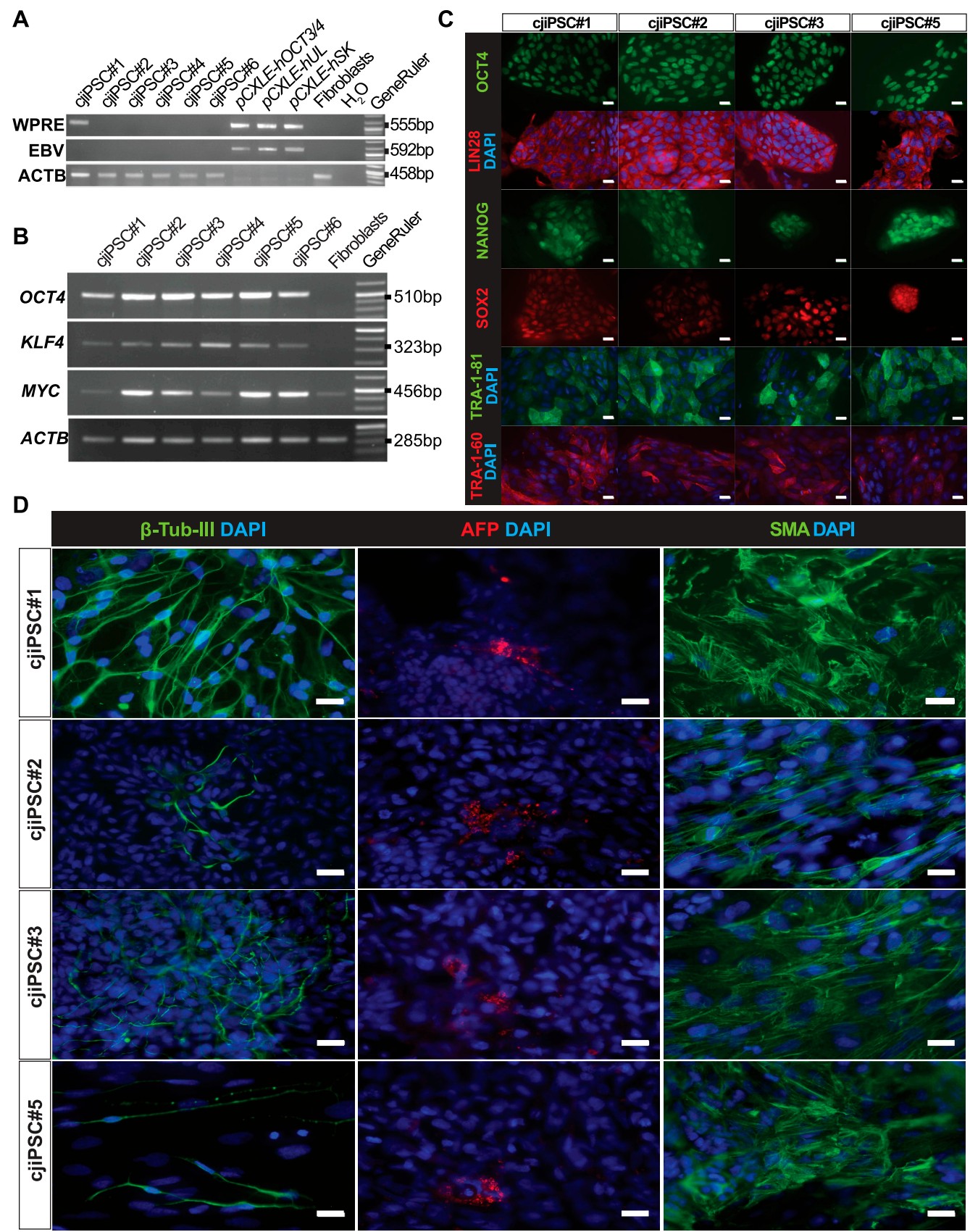

maintained in hEPSCM supplemented with TGFβ and FGF2, which promote pluripotency and self-renewal (James et al, 2005; Vallier et al, 2009; Chen et al, 2011; Gafni et al, 2013). cjiPSCs cultured in this medium, hereafter referred to as cjPSCM (marmoset PSC medium), homogenously express OCT4A, LIN28, and TRA-1-60 (Fig S3A), suggesting that pluripotency is maintained. To further characterise the pluripotent state of cjiPSCs cultured in TESR or cjPSCM, we performed RNA sequencing (RNA-seq). Differential gene expression analysis using DESeq2 ($Padj < 0.05$, $\log_2FoldChange > 2$) revealed 857 up- and 527 down-regulated genes in cjiPSCs cultured in cjPSCM as compared to TESR medium (Fig 4A). The expression of core pluripotency factors including OCT4A, NANOG, and SOX2 was unchanged. This is in contrast to the expression profile of hPSCs cultured in hEPSCM, where these factors are up-regulated compared with primed conditions (Gao et al, 2019). Also, we could not detect changes in the expression levels of DNA methyltransferases DNMT3A, DNMT3B, and DNMT1. However, many transcription factors associated with differentiation, such as WNT3A, PAX6, FGF9, HEY1, and TBX-T, were significantly down-regulated in cjiPSCs cultured in cjPSCM. Accordingly, gene ontology (GO) classification with down-regulated genes showed enrichment of terms associated with cellular differentiation, including *pattern specification process*, *regionalisation*, *embryonic organ development*, and *urogenital system development* (Fig S3B). In contrast, transcription factors considered to be associated with naïve pluripotency including *DPPA3*, *TFAP2C*, *KLF5*, *KLF4*, *GDF3*, and *ANPEP* were significantly up-regulated (Fig 4A) (Nakamura et al, 2016; Boroviak & Nichols, 2017; Bergmann et al, 2022). These results indicate that cjiPSCs require a species-specific cocktail of factors for expanded pluripotency, as cjPSCM does not fully induce the transcriptional changes associated with this stem cell state. Consistently, the media composition to induce expanded pluripotency is different for mouse, porcine, and hPSCs (Yang et al, 2017; Gao et al, 2019).

To further characterise the pluripotent state of cjiPSCs in cjPSCM, we compared the transcriptomes of cjiPSCs with available published datasets of hPSCs in naïve, formative, expanded potential and primed conditions (Gao et al, 2019; Kinoshita et al, 2021). Principal component analysis shows that hPSCs are separated across the pluripotency spectrum (Fig S3C). In particular, naïve hPSCs are distant from all other samples, whereas primed hPSCs are separated from formative and expanded pluripotent hPSCs along PC2. Interestingly, we observed a comparable shift of cjiPSCs cultured in cjPSCM from cjiPSCs in TESR medium. Similar to formative and expanded pluripotent hPSCs, cjiPSCs in cjPSCM express a subset of genes associated with naïve and formative pluripotency including ANPEP, PECAM1, POU5F1, NANOG, KLF5, DPPA3, NLRP7, DPPA3, KLF5, OTX2, and TFAP2C, as well as primed pluripotency including FAT3, THY1, and SOX11 (Fig 4B). However, there are also

notable differences including the up-regulation of SOX17 and FOXC1 in cjiPSCs. Taken together, these results indicate that cjPSCM induces gene expression changes in cjiPSCs that may lead to a partial reversion from primed to a formative-like state of pluripotency.

Next, we asked whether cjiPSCs cultured in cjPSCM gain competence to differentiate into pre-ME and cjPGCLCs. Indeed, immunofluorescence analysis of d4 EBs shows the expression and colocalisation of SOX17, AP2γ, NANOG, and BLIMP1 in a small number (6.9%, SEM: 1.51%) of cells (Fig 5A). In contrast, the definitive endoderm marker FOXA2 was only detected in a few SOX17 single-positive cells (Fig S3D). In addition, we empirically tested different conditions and found that the removal of ActA from the pre-ME differentiation medium apparently increased the number (9.9%, SEM: 2.65%) of cjPGCLCs (Fig 5B). Importantly, the differentiation of two additional cjiPSC lines (DPZ_cjiPSC#3 and DPZ_cjiPSC#5) in cjPSCM led to SOX17-BLIMP1 double-positive cells in d4 EBs (Fig S4A and B). Taken together, these results suggest that cjiPSCs cultured in feeder-free conditions with cjPSCM can be differentiated into pre-ME and induced into cjPGCLCs.

These results prompted us to test whether cjiPSCs in primed conditions with TESR medium, give rise to cjPGCLC-competent pre-ME if ActA is omitted during differentiation. Although immunofluorescence analysis of d4 EBs shows the expression of SOX17, AP2γ, and BLIMP1 in a low number of cells, the expression of these PGC markers was largely not colocalised (Fig S5A), which is reminiscent of EBs induced from pre-ME with ActA (Fig S2C). To characterise this further, we performed RNA-seq with pre-ME induced with or without ActA from cjiPSCs in TESR medium. Differential gene expression analysis using DESeq2 ($Padj < 0.05$, $\log_2FoldChange > 2$) showed significant up-regulation of 176 genes, many of which are associated with mesoderm development, including EOMES, SNAI1, GSC, HAND3, and WNT3 (Fig S5B and C). Consistently, GO classification with up-regulated genes shows enrichment for terms such as *muscle tissue development*, *pattern specification process*, *gastrulation*, and *mesoderm development* (Fig S5D). Considering that WNT signalling is required for PGC specification (Chen et al, 2017, 2019), it may be that the up-regulation of various WNT-associated genes upon the addition of ActA primes pre-ME from cjiPSCs in TESR medium towards the mesoderm fate. Taken together, these results demonstrate that cjiPSCs in a pluripotent state associated with cjPSCM in contrast to TESR medium give rise to cjPGCLC-competent pre-ME.

### cjPGCLCs express genes associated with the PGC fate

We asked whether cjPGCLCs up-regulate the germ cell–specific transcriptional programme, which entails the up-regulation of genes associated with pluripotency and the germ cell fate, whereas

---

**Figure 3. Reprogrammed cjiPSC lines are transgene-free and express pluripotency markers.**
**(A)** Agarose gel electrophoresis of PCR products. Two different primer combinations (WPRE and EBV), specific for two different regions conserved between the three episomes, were used to confirm the absence of reprogramming plasmids on the genomic DNA of the different cell populations. Episomes (pCXLE-hOCT3/4-shp53, pCXLE-hSK, and pCXLE-hUL) were used as a positive control, and fibroblasts and water as biological and technical negative controls, respectively. **(B)** Agarose gel electrophoresis of RT–qPCR-amplified products using the primers for *OCT4A*, *KLF4*, *c-MYC*, and *ACTB*. cDNAs of DPZ_cjiPSC#1-6 were analysed, and fibroblast cDNA was used as a control. **(C)** Immunofluorescence images of cjiPSCs (DPZ_cjiPSC#1, DPZ_cjiPSC#2, DPZ_cjiPSC#3, and DPZ_cjiPSC#5) stained for OCT4, LIN28, NANOG, SOX2, TRA-1-81, and TRA-1-60. **(D)** Immunofluorescence images of sections of cell aggregates formed by differentiated cjiPSC lines (DPZ_cjiPSC#1, DPZ_cjiPSC#2, DPZ_cjiPSC#3, and DPZ_cjiPSC#5). Sections were stained for β-Tub-III, AFP, and SMA. Scale bars, 20 μm.

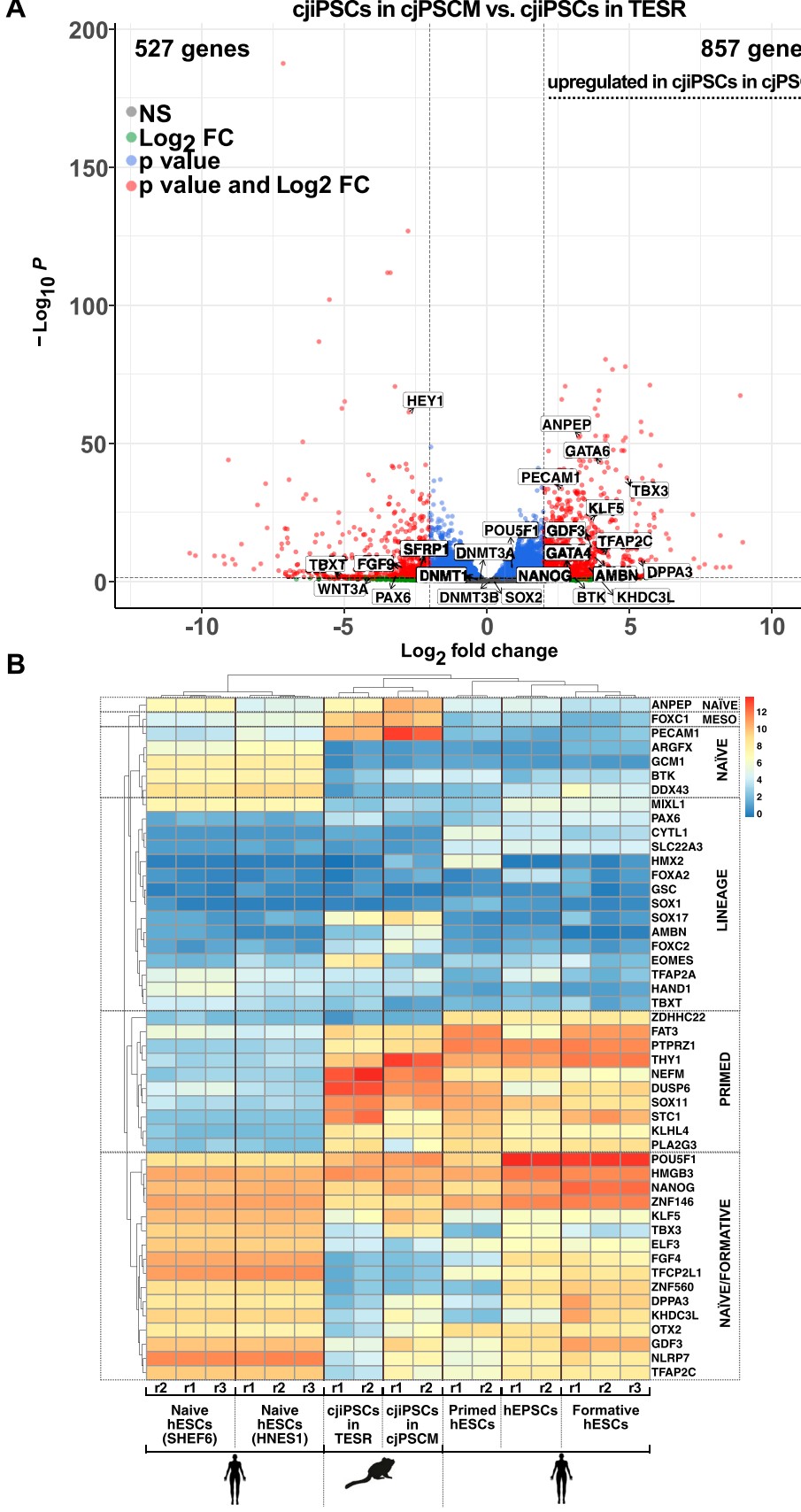

**Figure 4. cjiPSCs cultured in cjPSCM express naïve, formative, and primed pluripotency markers.**
**(A)** Volcano plot shows differential gene expression analysis of RNA-seq data for cjiPSCs cultured in cjPSCM compared with cjiPSCs in TESR medium. n = 2 biological independent replicates. **(B)** Heatmap showing expression levels of indicated genes using RNA-seq datasets for cjiPSCs cultured in TESR or cjPSCM compared with published datasets for hESCs cultured in naïve (E-MTAB-5114), formative (GSE131556), expanded (hEPSCs, E-MTAB-7253), and primed (E-MTAB-7253) pluripotent conditions. Scale: log$_2$(normalised counts + 1); r, replicate; MESO, mesoderm marker.

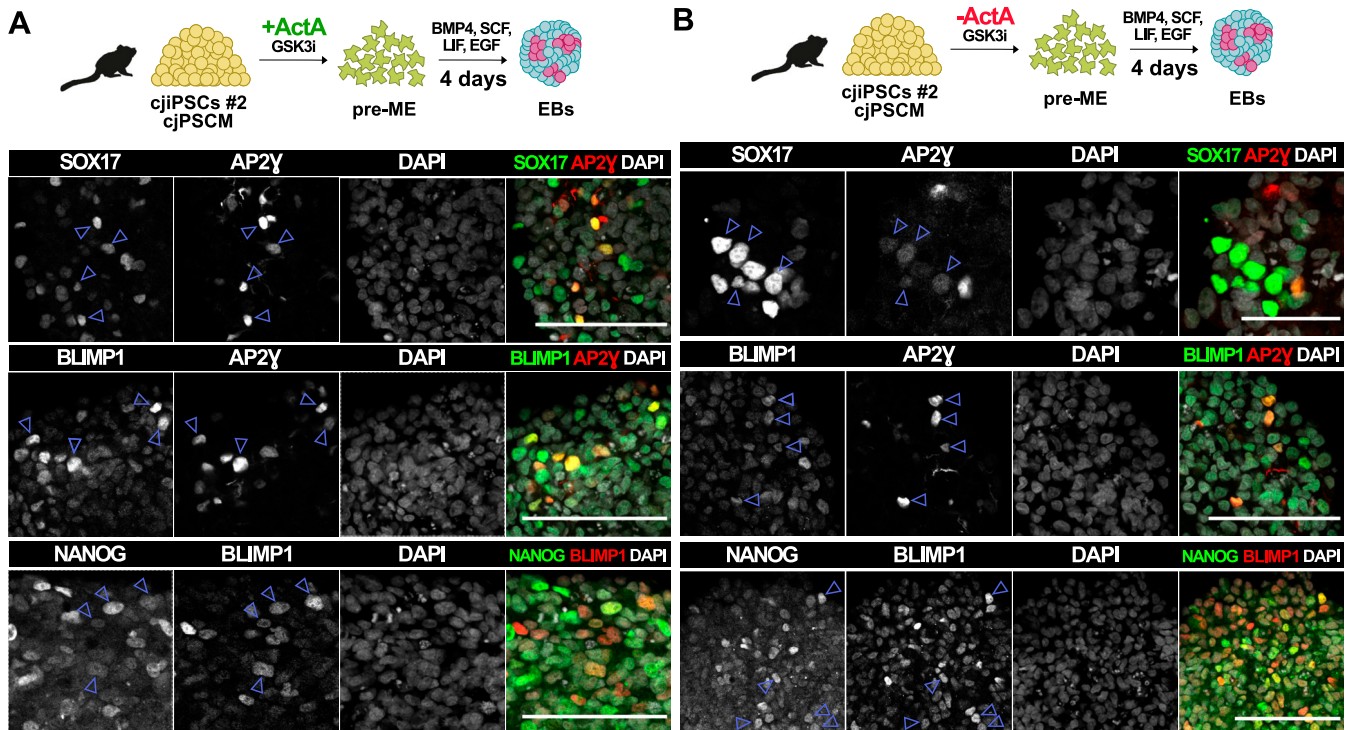

**Figure 5.   cjiPSCs cultured in cjPSCM give rise to cjPGCLCs.**
**(A, B)** Differentiation of cjiPSCs in cjPSCM into pre-ME, induced either with ActA (A) or without ActA (B), which was followed by cjPGCLC induction. IF images of resulting d4 EB sections stained for SOX17, AP2γ, BLIMP1, and NANOG. Scale bars, 100 μm.

genes associated with somatic fates are down-regulated (Kurimoto et al, 2008; Irie et al, 2015). To this end, we differentiated cjiPSCs in cjPSCM with or without ActA into pre-ME, which were then induced into cjPGCLCs. CjPGCLCs were isolated by FACS from d4 EBs after staining for the cell surface markers INTEGRINα6 (INTα6) and CXCR4 (Fig 6A–C), which are routinely used to sort primate PGCLCs (Sasaki et al, 2015; Kojima et al, 2017; Mitsunaga et al, 2017; Sakai et al, 2020; Seita et al, 2023).

RNA-seq analysis of pre-ME induced with ActA (pre-ME + ActA) resulted in significant up-regulation of differentiation-associated genes associated with GO terms such as *pattern specification process* and *mesoderm development* (Fig S6A). This includes *EOMES* and *WNT3A* (Fig S6B), which are involved in PGCLC specification in primates (Chen et al, 2017, 2019). Similarly, pre-ME induced without ActA (pre-ME – ActA) showed up-regulation of genes associated with *pattern specification process* and *BMP signaling pathway* (Fig S6C), which, however, did not include EOMES (Fig S6D). Instead, we noticed a pronounced transcriptional increase of TFAP2A (Fig S6D), which is required for the induction of the hPGC fate from progenitor cells (Chen et al, 2019; Castillo-Venzor et al, 2023). Direct comparison of pre-ME + ActA with pre-ME – ActA revealed 89 and 198 of up- and down-regulated genes (Padj < 0.05, log$_2$FoldChange > 2), respectively. Most notably, HOXA genes in pre-ME – ActA were significantly down-regulated (Fig S6E). These data show that the expression of somatic differentiation markers is up-regulated in pre-ME, which is more pronounced in pre-ME induced with ActA. Moreover, the up-regulation of TFAP2A in pre-ME – ActA might be an indicator of cjPGCLC competence.

Finally, we asked whether marmoset pre-ME retain the expression of subsets of genes associated with pluripotency, as it was shown for human pre-ME (Kobayashi et al, 2017; Tang et al, 2022; Alves-Lopes et al, 2023). Analysis of gene expression profiles in marmoset cjiPSCs and pre-ME compared with human ESCs and pre-ME using published RNA-seq datasets (Tang et al, 2022) confirmed that they share the expression of subsets of genes associated with naïve and primed pluripotency (Fig S7).

The INTα6-CXCR4 double-positive cjPGCLCs induced from pre-ME + ActA or pre-ME – ActA as compared to INTα6-CXCR4 double-negative somatic cells showed up-regulation of key PGCLC markers, including *SOX17, TFAP2C, PRDM1, EOMES, NR5A2, NANOG, OCT4 (POU5F1)*, and *DPPA3* (Fig 6D and E). Importantly, this was also the case for *NANOS3*, which is specifically expressed in the germline (Tsuda et al, 2003). Conversely, somatic genes including *HOXD4, HOXB6, MIXL1*, and *HAND1* were down-regulated. Also, the pluripotency factor *SOX2* was expressed but down-regulated, which is one of the key differences between rodent and primate PGC development (Irie et al, 2015; Sasaki et al, 2015; Sugawa et al, 2015).

Next, we compared the transcriptomes of cjPGCLCs with published datasets for mPGCLCs and human PGCLCs (hPGCLCs) (Sasaki et al, 2015; Tang et al, 2022). Pearson's correlations show that the transcriptomes of d4 cjPGCLCs cluster with d4 hPGCLCs (0.59–0.63) (Fig 7A). In contrast, both hPGCLC and cjPGCLC transcriptomes correlate less (0.25–0.41) with mPGCLCs at d4 or d6 of differentiation.

To characterise this further, we analysed the gene expression profile of various PGC markers between PGCLC samples,

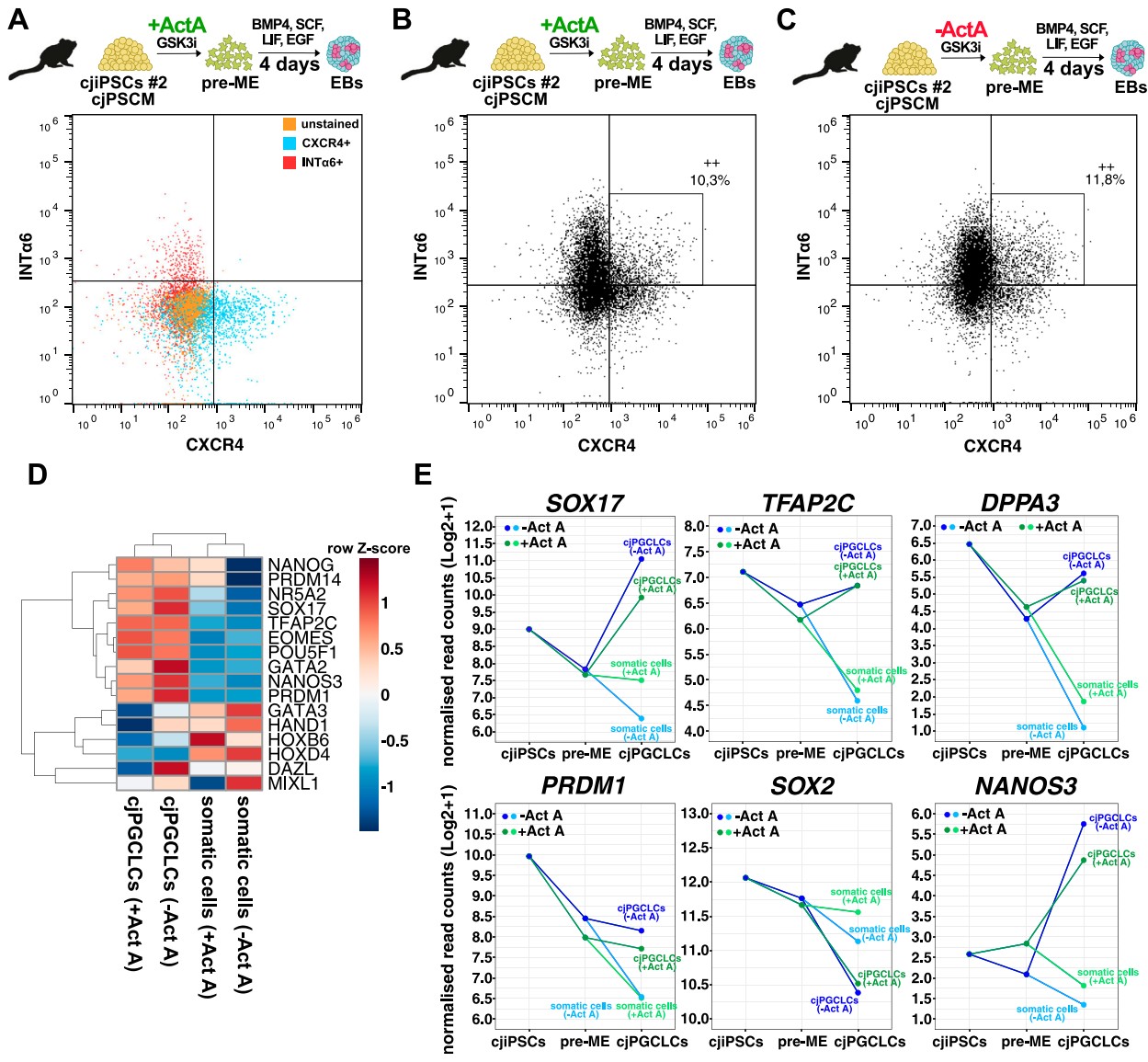

**Figure 6. cjPGCLCs up-regulate genes associated with the PGC fate.**
**(A)** Representative FACS analysis of d4 EBs: unstained control (orange), single-stained CXCR4 (blue), or INTα6 (red) control. **(B, C)** FACS analysis of d4 EBs stained for INTα6 and CXCR4. EBs were differentiated from cjiPSCs cultured in cjPSCM and further induced into pre-ME with Act A (pre-ME + ActA) (B) or without Act A (pre-ME – ActA) (C), which was followed by cjPGCLC induction. The number in % indicates INTα6/CXCR4 double-positive cells. **(D)** cjiPSCs in cjPSCM were differentiated into pre-ME with (+) or without (–) ActA, which were subsequently induced into cjPGCLCs. Heatmap shows relative gene expression levels based on RNA-seq data of indicated genes for sorted cjPGCLCs (INTα6/CXCR4 double-positive) and somatic cells (INTα6/CXCR4 double-negative). n = 2 biological independent replicates. **(E)** Expression levels (log₂(normalised counts + 1)) of indicated genes during differentiation.

including published datasets for embryonic human PGCs at week 7 of development (Tang et al, 2022). A large number of genes showed a similar expression profile in PGCLCs of all three species including the expression of POU5F1 (OCT4), NANOG, PRDM1, TFAP2C, and KIT (Fig 7B). Notably, PRDM14 could be only detected in hPGCLCs and mPGCLCs, indicating species-specific differences. Importantly, we also observed primate-specific up-regulation of PGC-associated genes including SOX17, KLF4, TFAP2A, GATA3, and TBX3 in hPGCLCs and cjPGCLCs as opposed to mPGCLCs. Taken together, these data suggest that cjiPSCs cultured in cjPSCM in feeder-free conditions can be

induced into cjPGCLCs with a transcriptional profile reminiscent of hPGCLCs.

## Discussion

The marmoset monkey has gained increasing attention in the field of germ cell research as a surrogate model to study early primate development. In this study, we further extend the dataset for in vivo cjPGC characterisation by immunostaining marmoset genital ridges at day E74 (equivalent to CS18). This analysis confirmed the

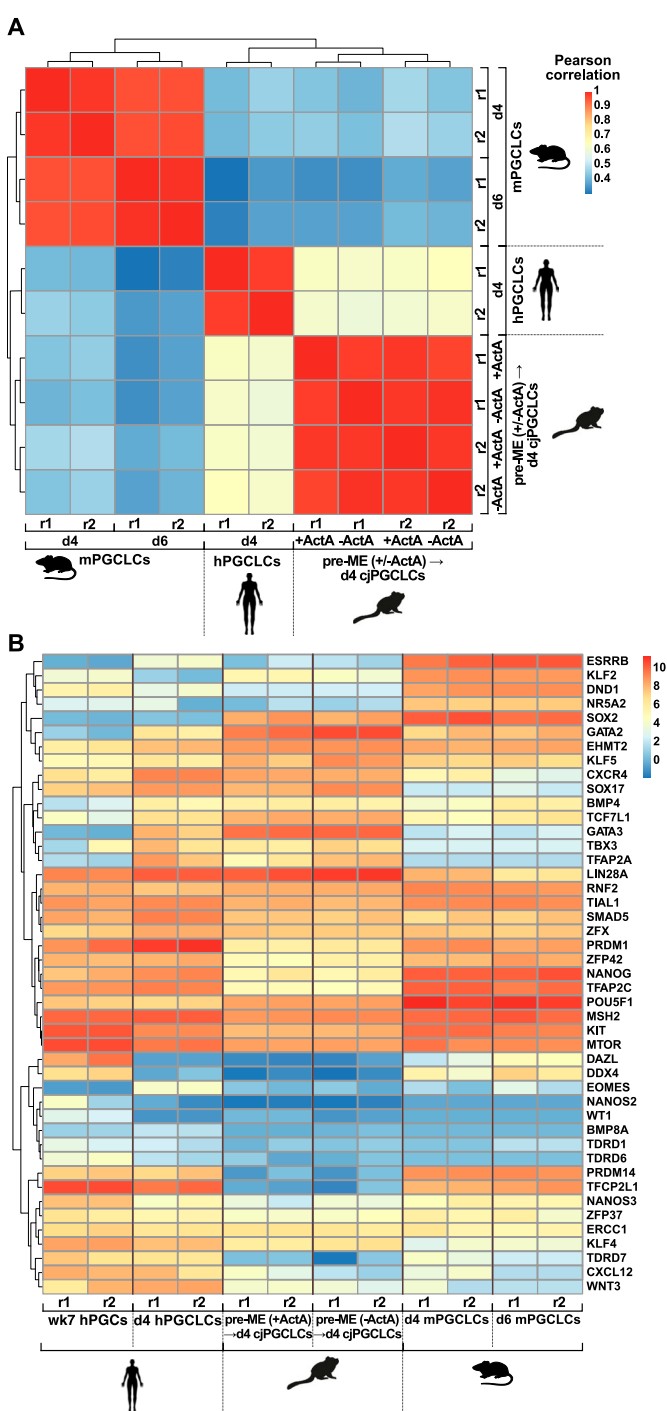

**Figure 7. Transcriptional programme of mPGCLCs, cjPGCLCs, and hPGCLCs.**
**(A)** Pearson's correlation of RNA-seq datasets for d4 cjPGCLCs induced from pre-ME (+/−ActA) and published data for d4/d6 mPGCLCs (GSE67259) and d4 hPGCLCs (GSE159654). **(B)** Heatmap showing expression levels of genes associated with PGC development in d4 cjPGCLCs, d4/d6 mPGCLCs, d4 hPGCLCs, and hPGCs (week 7, GSE159654). Scale: $\log_2$(normalised counts + 1), r = replicate.

presence of gonadal cjPGCs expressing AP2γ, BLIMP1, OCT4A, NANOG, and SOX17 but not SOX2 at the selected developmental time point (Fig 1), consistent with a previous study of cjPGCs at E50 (Seita et al, 2023). Thus, as a New World monkey, marmosets share

molecular key features of PGC developmental traits with humans, Old World monkeys, pigs, and rabbits (Irie et al, 2015; Sasaki et al, 2016; Kobayashi et al, 2017, 2021). Marmoset PSCs may therefore provide an attractive model for complete in vitro gametogenesis, which has only been achieved with mouse PSCs to date (Yoshino et al, 2021; Murakami et al, 2023).

We have generated a panel of transgene-free and feeder-free cjiPSCs using episomes for the transient expression of reprogramming factors (Okita et al, 2011). Most cjiPSC lines were transgene-free (except line #1) and could also be maintained in feeder-free conditions using a combination of UPPS medium and Geltrex for coating (Fig 2C), which we have previously used to successfully generate feeder-free human, macaque, and baboon iPSCs (Stauske et al, 2020). The UPPS formulation consists of StemMACS iPS-Brew XF as basal medium and the combination of CHIR and IWR-1, which activate and inhibit WNT signalling, respectively. WNT inhibition is required to maintain cyESCs and cjiPSCs on MEFs (Sakai et al, 2020; Seita et al, 2023). In our hands, the combination of CHIR and IWR-1 contributes to the stabilisation of NHP cell lines, which are per se more unstable in feeder-free culture than their human counterparts (Petkov et al, 2020; Stauske et al, 2020). Consistently, CHIR/IWR-1 can act synergistically to promote self-renewal of human ESCs and mouse epiblast stem cells by first activating (CHIR) and then stabilising (IWR-1) β-catenin in the cytoplasm (Kim et al, 2013). In addition, during the first passages of cjiPSC derivation, we also supplemented the medium with ActA and LIF. ActA has been shown to be part of the factors secreted by the feeder cells in MEF-dependent culture, and contributes to the maintenance of marmoset and human PSCs in an undifferentiated state at low concentrations (Beattie et al, 2005; James et al, 2005; Vallier et al, 2009; Nii et al, 2014). Interestingly, the combination of ActA and LIF promotes better morphology of the generated cjiPSC lines than ActA alone, which is in contrast to the culture conditions described for the maintenance of cjESCs (Nii et al, 2014). In addition, we show that cjiPSCs can also be cultured in TESR medium without feeders, which demonstrates the possibility of maintaining NHP PSCs in commercial medium formulations designed for human PSCs without the addition of supplements. This does not necessarily conflict with the versatility of UPPS medium as it provides suitable conditions for both stable and unstable cell lines. Further work will be required to dissect the similarities and differences between marmoset versus human pluripotency to understand the reasons for the preferred culture conditions for cjiPSCs.

Culture conditions determine the pluripotent state of PSCs, which also appears to determine the competence for the PGCLC fate. Our attempts to differentiate cjPGCLCs from primed cjiPSCs cultured in UPPS, E8, or TESR medium were unsuccessful (Fig S8). Considering that PGCs arise within extraembryonic amniotic cells in cynomolgus monkey embryos (Sasaki et al, 2016), we sought to adapt cjiPSCs to hEPSCM, which confers developmental competence to both embryonic and extraembryonic lineages in hESCs/hiPSCs (Gao et al, 2019). To this end, we found that hEPSCM was only suitable to maintain cjiPSC lines in an undifferentiated state when FGF2 and TGFβ were added (cjPSCM). Compared with cjiPSCs cultured in TESR medium, we did not observe an increase in the expression of core pluripotency factors, which is the case for

human ESCs in hEPSCM (Gao et al, 2019). However, cjPSCM induced the expression of naïve and formative pluripotency-associated markers, whereas many lineage-associated genes were repressed (Fig 4). This suggests that in addition to the similarities between various NHP and human PSCs, the marmoset presents species-specific aspects of pluripotency regulation that require further investigation.

Our established culture conditions allow cjiPSCs to differentiate into pre-ME and subsequently into cjPGCLCs (Figs 5A and B and S4), which express various PGC markers and down-regulate somatic genes (Fig 6D and E). Without the addition of ActA, cjiPSCs were not only able to differentiate into pre-ME, but the resulting cjPGCLC population showed a higher expression of PGC-associated markers compared with somatic cells (Fig 6D and E). Both ActA and TGFβ binding to receptors can induce a SMAD-dependent regulation of common target genes (James et al, 2005). Thus, the addition of TGFβ to cjPSCM may be sufficient to subsequently induce pre-ME, whereas the addition of ActA during pre-ME differentiation may prone the cells towards mesoderm rather than the PGC fate, which is consistent with the up-regulation of genes associated with the GO term *mesoderm development* (Fig S6A). Moreover, BMP4 signalling secreted by extraembryonic mesenchyme appears to play the dominant role in amnion formation in marmosets as opposed to ACTIVIN/NODAL and FGF (Bergmann et al, 2022). Considering that primate PGCs and amniotic cells may originate from the same precursors (Chen et al, 2019; Castillo-Venzor et al, 2023), we can assume that PGC formation is also independent of ACTIVIN/NODAL signalling. This provides an alternative explanation for why cjPGCLCs can be specified in the absence of ActA.

Recent studies have demonstrated the differentiation of cjPGCLC from cjiPSCs, which were maintained on feeder cells (Seita et al, 2023; Shono et al, 2023). Feeder cells secrete essential growth factors and cytokines that promote self-renewal of human PSCs, including TGFβ (Eiselleova et al, 2008; Villa-Diaz et al, 2013). Importantly, we have shown that cjPGCLCs can be induced from cjiPSCs cultured without feeders when maintained in the cjPSCM containing TGFβ.

Our study adds a panel of cjiPSCs in feeder-free and chemically defined conditions that can give rise to cjPGCLCs, which resemble early marmoset and human PGCs/PGCLCs. The defined cell culture conditions allow efficient and targeted analysis of the factors that control and modulate cell differentiation. With this study, we aim to contribute to the refinement of NHP iPSC technologies and to strengthen the role of the marmoset monkey as a surrogate model to study primate germ cell development and the production of functional gametes in vitro.

# Materials and Methods

## Animals and isolation of marmoset primary fibroblasts and PSCs

Marmoset fibroblasts used for reprogramming were obtained from one foetal and two postnatal marmosets. Marmoset foetal fibroblasts were extracted from foetuses at day 70–74 of gestation used in an unrelated project (Wolff et al, 2019) licensed by the Lower Saxony's State Office of Consumer Protection and Food Safety (Niedersächsisches Landesamt für Verbraucherschutz und Lebensmittelsicherheit; LAVES; license number 42502-04-16/2129). These fibroblasts were previously also used for the generation of iPSCs in another study (Petkov et al, 2020). Neonatal skin fibroblasts were extracted from one male and one female animal. The neonatal marmoset monkeys were obtained from the breeding colony of the German Primate Center. In captivity, marmosets sometimes give birth to triplets or even quadruplets. However, the mother is often able to feed and rear only two neonates, which is the normal litter size in free-living marmosets. For animal welfare reasons, the newborn, which receives an insufficient amount of milk from the mother, loses weight and must then be euthanised before it begins to suffer and would die. The neonatal samples used in this study were taken from such neonates from triplet or quadruplet births.

An HiPSC line (HPSI0114i-kolf_2) was obtained from HipSci (https://www.hipsci.org) and a cyESC line MF12 from the laboratory of Azim Surani (Gurdon Institute, Cambridge).

## Transfection of primary cells and reprogramming

Marmoset fibroblasts were isolated for outgrowths of small skin samples and cultured in cell culture plates coated with 0.1% gelatine (50-189-667FP; Thermo Fisher Scientific) in fibroblast medium (89 ml DMEM (10569010; Gibco), 10 ml FBS (A3160501; Gibco), 1 ml non-essential amino acids (NEAA, 11140-035, 100×; Gibco), and 0.1 mM 2-mercaptoethanol (31350010; Thermo Fisher Scientific)) supplemented with 10 ng/ml bFGF (100-18B; PeproTech) and penicillin–streptomycin (15140122; Thermo Fisher Scientific). After passage, two primary cell lines were propagated in fibroblast medium without antibiotics. The fibroblasts were transfected between passages 3 and 5 after isolation. Cultures with the highest proliferation potential were selected to increase reprogramming efficiency. A total of $1 \times 10^6$ cells were transfected using 4D-Nucleofector (Lonza) (Program CA137) with 6 µg plasmid DNA, containing the episomal vectors pCXLE-hSK (#27078; Addgene), pCXLE-hUL (#27080; Addgene), and pCXLE-hOCT3/4-shp53-F (plasmid #27077; Addgene). After nucleofection, the fibroblasts were maintained in Geltrex-coated dishes (A1413202; Thermo Fisher Scientific) in fibroblast medium supplemented with a pro-survival compound (ROCKi, 688000; Merck) for 1 d, and supplemented with 0.5 mM sodium butyrate (B5887; Sigma-Aldrich) for the next 6d. At d8, the medium was replaced with E8 medium (A1517001; Gibco) or UPPS medium (Table S1) with 0.5 mM sodium butyrate until d11. From d12, the transfected fibroblasts were maintained in Essential 8/UPPS medium alone. Putative iPSC colonies were manually picked from the primary plates and transferred to freshly Geltrex-coated plates with UPPS medium.

## Maintenance of cjiPSCs, cyESCs, and hiPSCs

CjiPSCs were cultured on Geltrex-coated (A1413202; Thermo Fisher Scientific) dishes under feeder-free conditions unless stated otherwise. CjiPSCs were established in the UPPS (Stauske et al, 2020) or alternatively in UPPS medium supplemented with 4 ng/ml Activin A (120-14P; PeproTech) and 10 ng/ml recombinant human LIF (300-05; PeproTech) (Table S1). After stabilisation, cjiPSCs were maintained in UPPS medium for expansion and characterisation.

For the generation of PGCLCs, cjiPSCs were transferred to modified UPPS medium (Petkov et al, 2020) as previously described. Briefly, cjiPSCs were cultured in StemMACS iPS-Brew XF medium (130-104-368; Miltenyi Biotec) supplemented with 1 μM IWR-1 (I0161-5MG; Sigma-Aldrich), 0.5 μM CHIR (Stem Cell Institute (SCI)), 10 μM forskolin (S2449; Selleckchem), and 1 ng/ml ActA (120-14P; Pepro-Tech). In addition, we tested the culture of the cjiPSCs and cyESCs in Essential 8 (E8) medium (A1517001; Thermo Fisher Scientific) with the addition of 5% KSR (10828010; Gibco) and 2.5 μM IWR1 on feeder cells. cjiPSCs were also differentiated into PGCLCs after adaptation to TeSR-E8 medium (05990; StemCell Technologies). Finally, to increase the competence, the cjiPSC lines were transferred to cjPSCM, which is derived from hEPSCM, but with the addition of 1 ng/ml TGFβ (100-21C; PeproTech) and 8 ng/ml FGF2 (SCI Facility, Cambridge).

CyESCs were cultured on vitronectin-coated (A14700; Gibco) dishes in E8 medium with the supplements as described above. hiPSCs were maintained on vitronectin-coated dishes in TeSR-E8 medium.

The medium was changed every day, and the cells were passaged every 3–5 d using Versene (15040033; Gibco) (cjiPSCs and hiPSCs) or 0.25% trypsin/EDTA (25200056; Gibco) (cyESCs). 10 μM Y-27632 2HCl (S1049; Selleckchem) was added for 1 d after passaging.

## Spontaneous differentiation in vitro

cjiPSCs were differentiated in vitro using the EB formation assay according to a previously published protocol (Stauske et al, 2020). After treatment with collagenase type IV, cell clumps derived from single colonies were cultured as a suspension in UPPS medium for 24 h, to allow complete aggregation. The UPPS medium was then replaced with differentiation medium (IMDM [12440053; Thermo Fisher Scientific], 20% FBS [16000044; Gibco], 1x MEM NEAA [11140-035; Gibco], and 450 μM 1-thioglycerol [M1753; Sigma-Aldrich]). After 8 d, the EBs were transferred to a cell culture plate with coverslips coated with 0.1% gelatine (∅ 25 mm; Thermo Fisher Scientific). After 25 d from the start of the protocol, coverslips containing embryoid body outgrowths were fixed and immunostained.

## PGCLC induction

The induction of marmoset, human, and cynomolgus monkey PGCLCs was performed as previously described (Irie et al, 2015; Kobayashi et al, 2017). Briefly, confluent cjiPSCs, cyESCs, or hiPSCs were washed with 1x PBS (20012-019; Gibco) once and then trypsinised with 0.25% trypsin/EDTA (25200056; Gibco). Cells were incubated at 37°C for 3 min and resuspended in MEF medium (DMEM/F-12 [21331-020; Gibco], 10% FBS [10270106; Gibco], 2 mM L-Glutamine [25030081; Gibco], 100 U/ml penicillin–streptomycin [15140122; Gibco]). The cells were passed through a 35-μm cell strainer (352235; Falcon) to remove the clumps, and the obtained suspension was centrifuged at 200g for 4 min. The cell pellet was resuspended in aRB27 medium (Advanced RPMI 1640 [12633012; Gibco], 1% B-27 supplement [17504001; Gibco], 1x MEM NEAA [11140-035; Gibco], 2 mM L-Glutamine [25030081; Gibco], and 100 U/ml penicillin–streptomycin [15140122; Gibco]) and subjected either to direct PGCLC induction or first to pre-induction towards pre-ME and then to PGCLC induction.

For pre-ME induction, dissociated cells were seeded on vitronectin-coated 12-well plates containing aRB27 medium (Advanced RPMI 1640 [12633012; Gibco], 1% B-27 supplement [17504001; Gibco], 1x MEM NEAA [11140-035; Gibco], 2 mM L-glutamine [25030081; Gibco], and 100 U/ml penicillin–streptomycin [15140122; Gibco]) with ActA (aRB27, 100 ng/ml ActA [SCI Facility, Cambridge], 3 μM CHIR [SCI facility, Cambridge], 10 μM Y-27632 2HCl [S1049; Selleckchem]) or without ActA (aRB27, 3 μM CHIR, 10 μM Y-27632 2HCl) at a density of 200,000 cells/well. hiPSCs and cyESCs were pre-induced for 12 h in the medium with ActA, whereas cjiPSCs were pre-induced for 24 h in the medium either with or without ActA.

For PGCLC induction, dissociated PSCs (direct PGCLC induction) or pre-ME cells were washed with PBS, trypsinised for 3 min at 37°C, resuspended in MEF, passed through the strainer, and centrifuged at 200g for 4 min. The cell pellet was resuspended in aRB27 medium, and after adjusting the cell concentration to 5,000 (h and cm) or 10,000 (cj) cells/100 μl/well, the corresponding volume of the cell suspension was added to PGCLC medium (aRB27, 500 ng/ml BMP4 [314-BP; R&D Systems], 10 ng/ml hLIF [SCI], 100 ng/ml SCF [455-MC-010; R&D Systems], 50 ng/ml mEGF [2028-EG-200; R&D Systems], and 10 μM Y-27632 2HCl [S1049; Selleckchem]). 100 μl of the cell suspension in PGCLC medium was plated into an ultra-low U-bottom 96-well plate (650979; Greiner or 7007; Costar), and resulting EBs were cultured for 2–6 d.

## Alkaline phosphatase staining

Alkaline phosphatase staining was performed with Leukocyte Alkaline Phosphatase Kit (85L1; Sigma-Aldrich), following the manufacturer's recommendations.

## Immunofluorescence staining, EB processing, and imaging

For cjiPSC characterisation, cells grown on glass coverslips were fixed in 4% (wt/vol) paraformaldehyde (AGR1026; Agar Scientific) in PBS for 20 min at RT and then washed three times with PBS. For blocking, 1% BSA (8076.2; Carl Roth) in PBS was used, and subsequently, the cells were incubated with primary antibodies (Table S2), diluted in 1% BSA plus 1% Triton X-100 (BP151-100; Thermo Fisher Scientific) in PBS at 4°C overnight. The cells were then washed three times with PBS and incubated with secondary antibodies (Table S2), also diluted in 1% BSA, for 1 h at 37°C. DNA staining was performed by incubating the cells with 1 μg/ml DAPI (D9542; Sigma-Aldrich) diluted in water for 10 min at RT. The cells were mounted with Vectashield (H-1000; Vector Laboratories) and analysed with a fluorescence confocal microscope (LSM 980; Zeiss).

For EB processing, the EBs were collected and washed with PBS. For fixation, EBs were incubated in 4% (wt/vol) paraformaldehyde for 20 min at RT. The paraformaldehyde was discarded, and EBs were washed three times in PBS. EBs were subjected to either 1-h or overnight incubation in 10% sucrose at 4°C, followed by 1-h incubation in 20% sucrose at 4°C. EBs were embedded in OCT (6478.2; Cell Path), incubated for 30 min at 4°C, and frozen at −80°C until further use. Marmoset E74 genital ridges were processed as

described above. For cryosectioning, a Leica cryostat was used, and 8-$\mu$m sections were collected on charged slides (J7800AMNT; Epredia), which were stored at –80°C until further use.

For immunofluorescence staining of EBs or genital ridges, slides were first washed three times for 5 min in PBS and then incubated in permeabilisation buffer (PBS, 1% BSA, 0.1% Triton X-100) for 30 min in a humidified chamber. The slides were incubated with primary antibodies overnight at 4°C. The slides were washed in PBS and incubated with secondary antibodies for 1 h at RT. The slides were washed in PBS, incubated with DAPI, and washed again before mounting with Vectashield. Imaging was performed using a Zeiss fluorescence confocal microscope (LSM 980) with Zeiss Plan-Apochromat 10x/0,45 M27, Zeiss Plan-Apochromat 20x/0,8 M27, Zeiss LC LCI Plan-Apochromat 25x/0.8 Imm Korr DIC M27, or Zeiss Plan-Apochromat 40x objectives.

### DNA/RNA isolation and PCR

Genomic DNA was extracted from snap-frozen cell pellets using DNeasy Blood and Tissue Kit (69556; QIAGEN). The absence of reprogramming plasmid DNA was demonstrated by PCR using primers specific for regions conserved between the three plasmids (Table S3). PCR was performed using Taq DNA Polymerase with Standard Taq buffer (M0273S; New England Biolabs).

Total RNA was extracted from snap-frozen cell pellets using NucleoSpin RNA Plus Kit (740984.50; Macherey-Nagel), Arcturus PicoPure RNA Isolation Kit (KIT0204; Thermo Fisher Scientific), or RNeasy Plus Micro Kit (74034; QIAGEN) according to the manufacturer's instructions. gDNA was removed by DNase I treatment (M0303S; New England Biolabs). Oligos were synthesised by IDT.

### FACS

FACS was performed as previously described with some minor modifications (Kobayashi et al, 2017). Briefly, primate EBs were harvested on day 4, washed with PBS, and incubated in 0.25% trypsin/EDTA in a thermomixer at 850 rpm at 37°C for 9–13 min. The reaction was stopped with 3% FBS in PBS, and the cell suspension was pipetted until it was dissociated into single cells and then passed through the strainer. The cells were centrifuged, and the cell pellet was resuspended in 3% FBS in PBS. Then, the dissociated cells were stained for CXCR4-APC (306510; BioLegend) and INT$\alpha$6-BV421 (313624; BioLegend) for 1 h in the dark on ice. Afterwards, the cells were washed in 3% FBS in PBS and subjected to FACS using SONY Cell Sorter SH800Z.

### RNA-seq

40 ng of RNA from cjiPSCs, pre-ME, and FACS-sorted cells was subjected to RNA-seq using NEBNext Ultra II RNA Library Prep Kit for Illumina (E7775; NEB) with NEBNext Poly(A) mRNA Magnetic Isolation Module (E7490; NEB). The library preparation was performed according to the manufacturer's instructions. Libraries were multiplexed and sequenced (paired-end 100) using a NovaSeq 6000 instrument.

### RNA-seq analysis

Quality checking of raw reads was performed using FastQC v0.11.8 and subsequently mapped to the *C. jacchus* reference genome assembly calJac4 using STAR 2.7.0a (Dobin et al, 2013). Aligned reads were assigned to gene annotations using HTSeq-count, version 0.11 (Anders et al, 2015). Read normalisation and differential analysis were performed using DESeq2 (Love et al, 2014). Genes were considered differentially expressed if they had an absolute log$_2$Fold-Change > 2 and an adjusted *P*-value of <0.05. Heatmaps were generated using the pheatmap R package (https://rdrr.io/cran/pheatmap/). GO enrichment analysis was performed using clusterProfiler (Yu et al, 2012). Coverage tracks requiring bigwig files were generated using deepTools v3.3.1 (Ramírez et al, 2016).

## Supplementary Information

## Data Availability

All data are available in the main text or the supplementary materials. Next-generation sequencing data are available using the GEO accession number GSE243324. cjiPSC lines are available upon request.

## Acknowledgements

We would like to thank Angelina Berenson, Nicole Umland, Ulrike Gödecke, Elif Toprak, and Simone Kalinowski for the excellent technical assistance and Kerstin Zaft for the administrative support. This work was supported by Sofja Kovalevskaja Award of the Humboldt Foundation to U Günesdogan and by the German Centre for Cardiovascular Research (DZHK 81Z3300232) and the German Primate Center-Leibniz Institute for Primate Research (financed by the Bundesrepublik Deutschland and the Bundesländer) to R Behr. I Rodriguez Polo was supported by a Walter Benjamin fellowship of the DFG and the Klaus Tschira Boost Fund. O Dovgusha was supported by the International Max Planck Research School for Genome Science. J Kurlovich and CRV Cruz were supported by the International Max Planck Research School for Molecular Biology.

### Author Contributions

J Kurlovich: conceptualisation, data curation, formal analysis, validation, investigation, visualisation, methodology, and writing—original draft, review, and editing.
I Rodriguez Polo: conceptualisation, data curation, formal analysis, validation, investigation, visualisation, methodology, and writing—original draft, review, and editing.
O Dovgusha: data curation, formal analysis, investigation, visualisation, and methodology.
Y Tereshchenko: investigation and methodology.
CRV Cruz: validation, investigation, and methodology.

R Behr: conceptualisation, resources, formal analysis, supervision, funding acquisition, investigation, project administration, and writing—review and editing.

U Günesdogan: conceptualisation, resources, data curation, formal analysis, supervision, funding acquisition, investigation, visualisation, project administration, and writing—original draft, review, and editing.

## Conflict of Interest Statement

The authors declare that they have no conflict of interest.

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
