## [Reviewer comments · Life Science Alliance]

Life Science Alliance

Generation of marmoset primordial germ cell-like cells under chemically defined conditions

Julia Kurlovich, Ignacio Rodriguez Polo, Oleksandr Dovgusha, Yuliia Tereshchenko, Carmela Rieline V. Cruz, and Rüdiger Behr and Ufuk Günesdogan

DOI: <https://doi.org/10.26508/lsa.202302371>

Corresponding author(s): Ufuk Günesdogan, University of Göttingen and Rüdiger Behr, German Primate Center - Leibniz Institute for Primate Research

Review Timeline:

Submission Date:	2023-09-13
Editorial Decision:	2023-10-11
Revision Received:	2024-01-30
Editorial Decision:	2024-02-23
Revision Received:	2024-03-01
Accepted:	2024-03-04

Transaction Report:

October 11, 2023

Re: Life Science Alliance manuscript #LSA-2023-02371-T

Dr. Ufuk Günesdogan
University of Göttingen
Department of Developmental Biology
Justus-von-Liebig Weg 11
Göttingen 37077
Germany

Dear Dr. Günesdogan,

Thank you for submitting your manuscript entitled "Generation of marmoset primordial germ cell-like cells under chemically defined conditions" to Life Science Alliance. The manuscript was assessed by expert reviewers, whose comments are appended to this letter. We invite you to submit a revised manuscript addressing the Reviewer comments.

Thank you for this interesting contribution to Life Science Alliance. We are looking forward to receiving your revised manuscript.

Sincerely,

B. MANUSCRIPT ORGANIZATION AND FORMATTING:

Reviewer #1 (Comments to the Authors (Required)):

In this manuscript, Kurlovich et al refine current methodologies to generate marmoset PGC-like cells. They first generate new feeder-free marmoset iPSC lines, which then they use as a starting point to improve conditions for PGC-like cell specification. The method, tools, and improvements described in the manuscript will be useful for the community. The manuscript is well written, the results are clear, and the conclusions are supported by the data. I support the publication of this manuscript once the following comments have been addressed:

1. The authors find out that removing Activin-A from the pre-induction step is favourable. However, this was done only using the expanded potential conditions. Is it possible to obtain PGC-like cells from the conventional primed cultures if Activin-A is removed during pre-mesoderm differentiation? Does Activin-A bias primed cells towards mesoderm?
2. The identity of the cells cultured under expanded potential conditions is not clear. The authors claim they are naïve-like, but Figure 4A shows the expression of endoderm markers. Could the authors compare the transcriptome of their cells to available datasets of human PSCs cultured under different regimes (i.e., naïve, formative, primed, expanded potential)?

Minor comments:

- Please indicate conditions of imaging (e.g., microscope and objectives used)
- Figures 4B and 4C should be quantified.

Reviewer #2 (Comments to the Authors (Required)):

Julia and her colleagues investigated the induction of primordial germ cell-like cells (PGCLCs) in marmosets. They observed that culturing cjiPSCs in cjPSCM, as opposed to other mediums, maintained a more naïve pluripotent stem cell state and ensured PGC competence for subsequent differentiation. Additionally, the removal of ActA from the preME differentiation medium appeared to increase the number of cjPGCLCs.

Comments:

- (1) The paper conducted RNA-Seq of cjiPSCs, pre-ME, and FACS-sorted cells, examining marker genes to illustrate cell identity. However, this doesn't fully support the conclusion of a "transcriptome similar to human PGCs/PGCLCs" as stated in the abstract. The authors should leverage the RNA-Seq data for a comprehensive transcriptomic comparison between marmoset, mouse, and human, given the availability of public data for the latter two.
- (2) For the cjPSCM, the authors proved that cjiPSCs in cjPSCM is more naïve compared to other medium. This is different from human since human starts in a primed state and it is the intermediate state that shows the dual property of naïve and primed properties (PMID: 31875561; PMID: 36640324). I suggested comparing the naïve gene expression levels in marmosets and humans in their differentiation processes including iPSC and PreME stages.
- (3) I strongly recommend the authors include a schematic differentiation figure that outlines medium selection, differentiation steps, and outputs, providing a clearer visual representation of the process.
- (4) Consider changing the word "upregulate" to "express" in the subtitle "cjPGCLCs upregulate the germ cell-specific transcriptional program." As the title suggests a program, should be defined as a gene list with a specific number of genes. The authors could define this gene list and use a gene enrichment score to demonstrate it effectively.
- (5) From page 6 please state clearly which one cjiPSCs (DPZ_cjiPSC#1-6) was used, and if there is any difference of cell lines in PGCLC differentiation efficiency? Similarly, for the test of the medium, did the medium have the same effect on all lines, or did the author test it only in one cell line?

Response to reviewers

Manuscript ID: LSA-2023-02371-T

Title: Generation of marmoset primordial germ cell-like cells under chemically defined conditions

We would like to sincerely thank the reviewers for their constructive comments, which helped to significantly improve our paper. Please find our point-by-point response (in blue) below. Changes in the text of the paper are highlighted in green.

Reviewer #1:

In this manuscript, Kurlovich et al refine current methodologies to generate marmoset PGC-like cells. They first generate new feeder-free marmoset iPSC lines, which then they use as a starting point to improve conditions for PGC-like cell specification. The method, tools, and improvements described in the manuscript will be useful for the community. The manuscript is well written, the results are clear, and the conclusions are supported by the data. I support the publication of this manuscript once the following comments have been addressed:

1. The authors find out that removing Activin-A from the pre-induction step is favourable. However, this was done only using the expanded potential conditions. Is it possible to obtain PGC-like cells from the conventional primed cultures if Activin-A is removed during pre-mesoderm differentiation?

To address this question, we have differentiated cjiPSCs cultured in conventional primed conditions (TESR) into precursors of mesendoderm (pre-ME) without Activin A (ActA) and subsequently into cjPGCLCs. Immunofluorescence analysis of embryoid bodies (EBs) shows that the PGC markers SOX17, AP2 γ and BLIMP1 are expressed in a low number of cells (Revision Fig. 1). However, their expression does not appear to colocalise, which is similar to the results obtained with the addition of ActA (Fig. S2C). These results suggest that cjiPSCs cultured in primed conditions do not efficiently differentiate into PGCLC-competent pre-ME, regardless of the presence/absence of ActA. We have included these data in the manuscript (Fig. S5A).

Revision Fig. 1: cjiPSCs cultured in TESR do not give rise to cjPGCLCs. (A, B) cjiPSCs cultured in TESR medium were pre-induced towards pre-ME with (A) or without (B) ActA (+/-ActA), followed by cjPGCLC

differentiation. Immunofluorescence images of day 4 (d4) EB sections stained for SOX17, AP2γ, BLIMP1 and/or NANOG. Scale bars, 100 μm.

Does Activin-A bias primed cells towards mesoderm?

We have now performed RNA-seq with pre-ME differentiated with (pre-ME+ActA) or without ActA (pre-ME-ActA) from cjiPSCs in primed conditions (TESR). Differential gene expression analysis (DESeq2; padj < 0.05, log2foldchange > 2) shows significant upregulation of 176 genes in pre-ME+ActA compared to pre-ME-ActA (**Revision Fig. 2A**). As predicted by the reviewer, many of these genes are associated with mesoderm development, including EOMES, SNAI1, GSC, HAND3, WNT3 (**Revision Fig. 2B**). Consistently, gene ontology (GO) classification with upregulated genes shows enrichment for terms such as *muscle tissue development*, *pattern specification process*, *gastrulation*, and *mesoderm development* (**Revision Fig. 2C**). Taken together, these results indicate that ActA primes pre-ME for mesoderm fate during differentiation from cjiPSCs cultured in conventional primed conditions. We have included this data in (**Fig. S5B-C**).

Revision Fig. 2: Activin A primes pre-ME differentiated from primed cjiPSCs for mesoderm fate. (A) Differential gene expression analysis (DESeq2) of RNA-seq datasets of pre-ME differentiated with or without ActA (+/-ActA) from cjiPSCs cultured in TESR. (B) Heatmap showing expression levels of genes associated with mesoderm development in pre-ME. r: replicate. (C) GO term enrichment for genes significantly upregulated in pre-ME -ActA compared to pre-ME +ActA.

2. The identity of the cells cultured under expanded potential conditions is not clear. The authors claim they are naïve-like, but Figure 4A shows the expression of endoderm markers. Could the authors compare the transcriptome of their cells to available datasets of human PSCs cultured under different regimes (i.e., naïve, formative, primed, expanded potential)?

We agree with the reviewer that this conclusion needs to be revised. Indeed, we observe both the upregulation of some markers of naïve pluripotency and somatic differentiation in cjiPSCs cultured in cjPSCM versus TESR. As suggested by the reviewer, we compared the transcriptomes of our condition to available published datasets of human pluripotent stem cells (hPSCs) in naïve (E-MTAB-5114), formative (GSE131556), expanded potential (E-MTAB-7253) and primed conditions (E-MTAB-7253). First, we performed global transcriptome comparisons using principal component analysis (PCA; **Revision Fig. 3A**). hPSC samples are separated across the pluripotency spectrum. In particular, naïve hPSCs are distant from all other samples, whereas primed hPSCs are separated from formative and expanded pluripotent hPSCs along PC2. Interestingly, we observed a comparable shift of cjiPSCs cultured in cjPSCM from cjiPSCs in TESR. However, it should be noted that cjiPSCs are generally quite distant from hPSC samples, which is likely due to technical differences between RNA-seq datasets from different studies, but also species-specific differences.

Revision Fig. 3: Expression profile of cjiPSCs in cjPSCM compared to hESCs in different pluripotent states. (A) PCA with a defined set of pluripotency-associated genes using RNA-seq datasets for cjiPSCs cultured in TESR or cjPSCM compared to published datasets for hESCs cultured in naïve (E-MTAB-5114), formative (GSE131556), expanded (hEPSCs, E-MTAB-7253) and primed (E-MTAB-7253) pluripotent conditions. (B) Heatmap showing the expression levels of indicated genes. Scale: $\text{Log}_2(\text{normalised counts}+1)$; MESO = mesoderm marker.

To analyse this further, we compared the gene expression profile of markers of pluripotency and differentiation between marmoset and hPSC samples. Interestingly, similar to formative and expanded pluripotent hPSCs, cjPSCs in cjPSCM upregulate a subset of genes associated with naïve and formative pluripotency including ANPEP, PECAM1, POU5F1, NANOG, KLF5, DPPA3, NLRP7, DPPA3, KLF5, OTX2 and TFAP2C as well as primed pluripotency including FAT3, THY1, and SOX11 (**Revision Fig. 3B**). However, there are also notable differences including the upregulation of SOX17 and FOXC1 in cjiPSCs. Taken together, these results indicate that cjPSCM induces gene expression changes in cjiPSCs that may lead to a partial reversion from primed to a formative-like pluripotent state. We have included this analysis in the manuscript (**Fig. 4B and Fig. S3C**).

Minor comments:

- Please indicate conditions of imaging (e.g., microscope and objectives used)

We have included information on imaging in the Methods section.

- Figures 4B and 4C should be quantified.

We have quantified the immunofluorescence images shown in Fig. 4B/C, which shows 6.9% (+/-1.51%) cjPGCLCs induced from pre-ME+ActA as compared to 9.9 (+/-2.65%) cjPGCLCs induced from pre-ME-ActA. We have included this information in the manuscript.

Reviewer #2 (Comments to the Authors (Required)):

Julia and her colleagues investigated the induction of primordial germ cell-like cells (PGCLCs) in marmosets. They observed that culturing cjiPSCs in cjPSCM, as opposed to other mediums, maintained a more naïve pluripotent stem cell state and ensured PGC competence for subsequent differentiation. Additionally, the removal of ActA from the preME differentiation medium appeared to increase the number of cjPGCLCs.

Comments:

(1) The paper conducted RNA-Seq of cjiPSCs, pre-ME, and FACS-sorted cells, examining marker genes to illustrate cell identity. However, this doesn't fully support the conclusion of a "transcriptome similar to human PGCs/PGCLCs" as stated in the abstract. The authors should leverage the RNA-Seq data for a comprehensive transcriptomic comparison between marmoset, mouse, and human, given the availability of public data for the latter two.

Following the recommendation of the reviewer, we compared the transcriptomes of mouse PGCLCs (mPGCLCs; GSE67259), human PGCLCs (hPGCLCs, GSE159654), and cjPGCLCs (this study). Pearson correlations show that the transcriptomes of d4 cjPGCLCs correlate and cluster with d4 hPGCLCs (0.59-0.63) (**Revision Fig. 4A**). In contrast, both hPGCLC and cjPGCLC transcriptomes correlate less (0.25-0.41) with mPGCLCs at d4 or d6 of differentiation.

Next, we compared the gene expression profile of a comprehensive list of PGC markers between PGCLC samples, including embryonic human PGCs (hPGCs, week7, GSE159654) (**Revision Fig. 4B**). A large number of genes show a similar expression profile in all three species including the expression of POU5f1 (OCT4), NANOG, PRDM1, TFAP2C, and KIT. Notably, PRDM14 could be only detected in hPGCLCs and mPGCLCs, indicating species-specific differences. Importantly, we also observed primate-specific upregulation of PGC-associated genes including SOX17, KLF4, TFAP2A, GATA3, and TBX3 in hPGCLCs and

cjPGCLCs as opposed to mPGCLCs. Taken together, these data suggest that the transcriptional profile of cjPGCLCs is reminiscent of that of hPGCLCs. We have included this analysis in the manuscript (Fig. 7).

Revision Fig. 4: The transcriptional programme of mPGCLCs, cjPGCLCs and hPGCLCs. (A) Pearson correlation of RNA-seq datasets for d4 cjPGCLCs induced from pre-ME (+/-ActA) and published data for d4/d6 mPGCLCs and d4 hPGCLCs. (B) Heatmap showing expression levels of genes associated with PGC development. Scale: Log2(normalised counts+1).

(2) For the cjPSCM, the authors proved that cjiPSCs in cjPSCM is more naïve compared to other medium. This is different from human since human starts in a primed state and it is the intermediate state that shows the dual property of naïve and primed properties (PMID: 31875561; PMID: 36640324). I suggested comparing the naïve gene expression levels in marmosets and humans in their differentiation processes including iPSC and PreME stages.

Please note that we have revised our conclusion regarding the pluripotent state of cjiPSCs cultured in cjPSCM (please see Reviewer#1 point2 above).

Following the reviewer's suggestion, we analysed the expression levels of genes associated with naïve and primed pluripotency using published RNA-seq datasets for primed hESCs (GSE159654) and human pre-ME (GSE159654) and our datasets for cjiPSCs and marmoset pre-ME. Indeed, human and marmoset pre-ME share the expression of subsets of genes associated with naïve and primed pluripotency. This is consistent with the aforementioned previous study (PMID: 36640324¹), which shows by PCA that the global transcriptome of human pre-ME is close to that of primed hESCs (Fig. 1H in Alves-Lopes et al., 2023¹). Furthermore, we observe an upregulation of TFAP2A in marmoset pre-ME -ActA, which is associated with the developmental trajectory of hPGCLC fate, as shown by Chen et al. 2019 (PMID: 31875561²). Taken together, our data suggest that marmoset pre-ME harbour gene

expression features of naïve and primed pluripotency. We have included this analysis in the manuscript (**Fig. S7**).

Revision Fig. 5: Expression profile of genes associated with pluripotency in pre-ME. Heatmap showing expression levels of genes associated with naïve and primed pluripotency. RNA-seq datasets are from primed hESCs (GSE159654), human pre-ME ((h)pre-ME, GSE159654), cjiPSCs (this study) and marmoset pre-ME ((cj)pre-ME, this study). Scale: $\text{Log}_2(\text{normalised counts}+1)$.

(3) I strongly recommend the authors include a schematic differentiation figure that outlines medium selection, differentiation steps, and outputs, providing a clearer visual representation of the process.

We have included schematics of the differentiation approaches in the relevant figures and summarised this in **Fig. S8**.

(4) Consider changing the word "upregulate" to "express" in the subtitle "cjPGCLCs upregulate the germ cell-specific transcriptional program." As the title suggests a program, should be defined as a gene list with a specific number of genes. The authors could define this gene list and use a gene enrichment score to demonstrate it effectively.

Following the reviewer's recommendation, we changed the subtitle to '**cjPGCLCs express genes associated with PGC fate**'. We also performed gene set enrichment analysis (GSEA) with a defined set of PGC genes between cjPGCLCs and somatic cells, which confirmed a significant enrichment of PGC genes in cjPGCLCs (**Revision Fig. 6**). However, we have decided not to include this analysis in the manuscript, since we think that the newly added analysis (**Revision Fig. 4**) already shows the expression levels of individual PGC genes in cjPGCLCs compared to human and mouse PGCLCs.

Revision Fig. 6: Gene set enrichment for genes associated with PGCs. Gene set enrichment analysis (GSEA) shows significant enrichment of a defined set of PGC genes in d4 cjPGCLCs induced from pre-ME (+/-ActA) compared to somatic cells.

(5) From page 6 please state clearly which one cjiPSCs (DPZ_cjiPSC#1-6) was used, and if there is any difference of cell lines in PGCLC differentiation efficiency? Similarly, for the test of the medium, did the medium have the same effect on all lines, or did the author test it only in one cell line?

The in-depth characterisation was performed with cjPGCLCs induced from one cell line (DPZ_cjiPSC#2). Initially, we tested the direct induction of cjPGCLCs from two other cjiPSC cell lines (DPZ_cjiPSC#3, DPZ_cjiPSC#5) cultured in UPPS conditions (**Revision Fig. 7A**). We observed the induction of AP2 γ -BLIMP1 clusters at d2 of differentiation. However, at d6 of differentiation, only a few BLIMP1-positive cells remained, which appeared to be mutually exclusive with SOX17-expressing cells. This suggests that the cells adopted different cell fates during the time course of differentiation, which is consistent with the results obtained for DPZ_cjiPSC#2.

We also adapted the culture of DPZ_cjiPSC#3 and DPZ_cjiPSC#5 to cjPSCM. We then differentiated cjiPSC lines into pre-ME with or without Activin A and induced cjPGCLCs, resulting in clusters of SOX17-expressing cells (**Revision Fig. 7B**). In addition, a subset of these cells was co-expressing BLIMP1, which appeared to be more efficient in conditions where pre-ME was differentiated without Activin A. Thus, these results suggest that cjPSCM can be used to culture at least three independent cjiPSC lines for subsequent cjPGCLC differentiation. We have added this data (**Fig. S1 and S4**) and clarified the information about the cell lines in the manuscript.

Revision Fig. 7: Induction of cjPGCLCs from other cjiPSC lines. A, B DPZ_cjiPSC#2, #3 and #5 cultured in UPPS medium were directly induced into cjPGCLCs. Immunofluorescence images of d2 (**A**) or d6 (**B**) EB sections stained for SOX17, AP2 γ , and/or BLIMP1. **C, D** DPZ_cjiPSC#3 and #5 cultured in cjiPSCM were pre-induced into pre-ME with (**C**) or without (**D**) ActA (+/-ActA) and then into cjPGCLCs. Immunofluorescence images of d4 EB sections stained for SOX17 and BLIMP1. Scale bars, 100 μ m.

References for 'Response to Reviewers'

1. Alves-Lopes, J. P. *et al.* Specification of human germ cell fate with enhanced progression capability supported by hindgut organoids. *Cell Rep* **42**, 111907 (2023).
2. Chen, D. *et al.* Human Primordial Germ Cells Are Specified from Lineage-Primed Progenitors. *Cell Reports* **29**, 4568-4582.e5 (2019).

February 23, 2024

RE: Life Science Alliance Manuscript #LSA-2023-02371-TR

Dr. Ufuk Günesdogan
University of Göttingen
Department of Developmental Biology
Justus-von-Liebig Weg 11
Göttingen 37077
Germany

Dear Dr. Günesdogan,

Thank you for submitting your revised manuscript entitled "Generation of marmoset primordial germ cell-like cells under chemically defined conditions". We would be happy to publish your paper in Life Science Alliance pending final revisions necessary to meet our formatting guidelines.

- please be sure that the authorship listing and order is correct, and that they match in your manuscript and our system
- Please add ORCID ID for secondary corresponding author--you should have received instructions on how to do so
- Please upload each supplementary table separately in .docx file format
- Please move figure and table legends at the end of manuscript file after the 'References section'.
- Please label each section as: "Figure legends", "Supplementary figure legends" and "Supplementary Table legends"
- Please add a callout for Figure S4 both sections A and B to your main manuscript text
- GEO accession GSE243324 should be made publicly accessible at this time, and the Reviewer token can be removed from the Data Availability statement

Figure checks:

- What are the dashed in Figure 3A meant for? Are these to allow for easier reading of the columns, or do they indicate splices in the blot? Either way, please indicate what the dashes are in the legend.

A. FINAL FILES:

B. MANUSCRIPT ORGANIZATION AND FORMATTING:

Sincerely,

Reviewer #1 (Comments to the Authors (Required)):

The authors have satisfactorily addressed my comments. I support the publication of this manuscript.

March 4, 2024

RE: Life Science Alliance Manuscript #LSA-2023-02371-TRR

Dr. Ufuk Günesdogan
University of Göttingen
Department of Developmental Biology
Justus-von-Liebig Weg 11
Göttingen 37077
Germany

Dear Dr. Günesdogan,

Thank you for submitting your Methods entitled "Generation of marmoset primordial germ cell-like cells under chemically defined conditions". It is a pleasure to let you know that your manuscript is now accepted for publication in Life Science Alliance. Congratulations on this interesting work.

DISTRIBUTION OF MATERIALS:

Again, congratulations on a very nice paper. I hope you found the review process to be constructive and are pleased with how the manuscript was handled editorially. We look forward to future exciting submissions from your lab.

Sincerely,
